# Gold Nanoparticles: Can They Be the Next Magic Bullet for Multidrug-Resistant Bacteria?

**DOI:** 10.3390/nano11020312

**Published:** 2021-01-26

**Authors:** Mohammad Okkeh, Nora Bloise, Elisa Restivo, Lorenzo De Vita, Piersandro Pallavicini, Livia Visai

**Affiliations:** 1Center for Health Technologies (CHT), Department of Molecular Medicine, INSTM UdR of Pavia, University of Pavia, Viale Taramelli 3/b, 27100 Pavia, Italy; mohammadkhalil.okkeh01@universitadipavia.it (M.O.); nora.bloise@unipv.it (N.B.); elisa.restivo01@universitadipavia.it (E.R.); 2Medicina Clinica-Specialistica, UOR5 Laboratorio di Nanotecnologie, ICS Maugeri, IRCCS, Via S. Boezio 28, 27100 Pavia, Italy; 3Department of Chemistry, University of Pavia, Viale Taramelli 12, 27100 Pavia, Italy; lorenzo.devita01@universitadipavia.it (L.D.V.); piersandro.pallavicini@unipv.it (P.P.)

**Keywords:** antibacterial gold nanoparticles, photo-thermal and photo-dynamic therapy, nanomedicine, multidrug resistant bacteria

## Abstract

In 2017 the World Health Organization (WHO) announced a list of the 12 multidrug-resistant (MDR) families of bacteria that pose the greatest threat to human health, and recommended that new measures should be taken to promote the development of new therapies against these superbugs. Few antibiotics have been developed in the last two decades. Part of this slow progression can be attributed to the surge in the resistance acquired by bacteria, which is holding back pharma companies from taking the risk to invest in new antibiotic entities. With limited antibiotic options and an escalating bacterial resistance there is an urgent need to explore alternative ways of meeting this global challenge. The field of medical nanotechnology has emerged as an innovative and a powerful tool for treating some of the most complicated health conditions. Different inorganic nanomaterials including gold, silver, and others have showed potential antibacterial efficacies. Interestingly, gold nanoparticles (AuNPs) have gained specific attention, due to their biocompatibility, ease of surface functionalization, and their optical properties. In this review, we will focus on the latest research, done in the field of antibacterial gold nanoparticles; by discussing the mechanisms of action, antibacterial efficacies, and future implementations of these innovative antibacterial systems.

## 1. Introduction

Bacterial resistance, one of the biggest threats to human health in the 21st century, is the ability of bacterial cells to resist one or more types of antibiotics [1]. Infections like pneumonia and tuberculosis are becoming harder to treat because the available antibiotics are becoming less effective due to rising bacterial resistance [2,3]. The decline in the development of new antibiotics, along with their overuse and misuse, has made the situation even worse. All this is leading to protracted hospital stays, higher medical costs, and a rise in mortality [4].

Although many bacteria are still susceptible to the majority of antimicrobial agents available, a specific group of bacteria can escape the bactericidal action of many antibiotics. This small group consists of *Enterococcus faecium, Staphylococcus aureus, Klebsiella pneumonia, Acinetobacter baumanni, Pseudomonas aeruginosa,* and the *Enterobacter* species, and these bacteria are referred to as “the ESKAPE” pathogens [5]. These pathogens are essential for two reasons; they are, first, the causative agents for the majority of nosocomial infections and, second, they are considered as a model of pathogenesis, transmission, and resistance. Once we learn how to control these microorganisms the same strategies could be applied to other species that attempt to take their place [5].

### 1.1. Onset of Bacterial Resistance to Antibiotics and its Mechanism of Action

Bacterial resistance to antibiotics began as early as the 1950s, when penicillin resistance became a health concern [6]. The issue was temporarily solved with the introduction of beta-lactam antibiotics [6,7] but unfortunately this did not last long as the first case of methicillin-resistant *Staphylococcus aureus* (MRSA) was reported in the United Kingdom as early as 1961 [7]. From the late 1960s to the early 1980s more antibiotics were introduced to the market, but resistance to these drugs hindered their potential. This sadly led to the drying up of the antibiotic pipeline and fewer new antibiotics were subsequently introduced [6]. This factor—along with the misuse, inappropriate prescribing, extensive agricultural use of antibiotics, and regulatory barriers—has led to what is known as the antibiotic resistance crisis [8].

Antibiotic-resistance can be developed by bacteria by three main mechanisms, as illustrated in Figure 1: (i) The inhibition of the entry of antibiotic molecules to their target in the bacterial cells by either decreased permeability or increased efflux. Gram-negative bacteria are inherently less permeable to many antibiotics as their outer membrane is selectively permeable. Additionally, the down-regulation of porins, or their replacement by other more selective ones, hinders the access of antibiotics into the bacterial cells [9,10]. (ii) Altering antibiotic targets by mutation or posttranslational modification or protection. The modification of the target part by mutation prevents the efficient binding of antibiotic molecules, which allows the bacteria to function normally and thus acquire resistance against the drug. On the other hand, protection or modification of the target structure can happen without any mutations in the encoding genes, and this may also lead to antibiotic resistance [11]. (iii) Resistance by directly acting upon the antibiotics by either hydrolysis or transferring of a chemical group; this is considered the primary mechanism of antibiotic-resistance. One classic example is the penicillinase enzyme degrading penicillin antibiotics. Altering antibiotics by the addition of chemical groups (acyl, phosphate, nucleotidyl, and ribitoyl, etc.) to the active sites is another behavior employed by bacteria to prevent antibiotics from binding to the target as a result of steric hindrance [12,13].

Another big challenge that faces healthcare systems in dealing with bacterial resistance is the formation of biofilms. Biofilms are functional aggregates of sessile microorganisms encased within a self-generated extracellular polymeric matrix composed of polysaccharides, proteins, and DNA [14,15]. The term “aggregate” is used because most cells in multilayered biofilms establish cell-to-cell contact, either in surface-attached biofilms or in flocs, which are mobile biofilms. Through intercellular interactions, both social and physical, together with the properties of the matrix, the biofilms differ considerably from free-living bacterial cells [15]. They possess different architectural, phenotypic, and biochemical properties that give them superiority over their planktonic counterparts in terms of pathogenicity and resistance to antimicrobial agents [14,16,17]. Due to these facts the search for innovative solutions to bacterial resistance has become a necessity on a global level.

Bacterial Resistance in Numbers according to Recent Epidemiological Data:

According to the latest annual epidemiological report of antimicrobial resistance in the European Union, EU/EEA (EARS-Net) in 2019 [18], statistics regarding the most commonly reported bacterial species were as indicated in Table 1.: *Escherichia coli* (44.2%), followed by *S. aureus* (20.6%), *K. pneumonia* (11.3%), *Enterococcus faecalis* (6.8%), *P. aeruginosa* (5.6%), *Streptococcus pneumoniae* (5.3%), *E. faecium* (4.5%), and *Acinetobacter* species (1.7%). The report showed that in 2019, more than half of the *E. coli* isolates reported to EARS-Net, and more than a third of the *K. pneumoniae* isolates, were resistant to at least one antimicrobial group under surveillance, and combined resistance to several antimicrobial groups was frequent.

For *S. aureus*, a decline in the percentage of meticillin-resistant *S. aureus* (i.e., MRSA) isolates reported in previous years continued, where it decreased from a percentage of 19% in 2015 to 15.5% in 2019. However, despite this good news, MRSA levels are still high in several countries, and combined resistance to another antimicrobial groups was common. *S. aureus* is also one of the most frequent causes of blood-borne infections, causing a high burden in terms of morbidity and mortality [19]. A worrying finding in this recent report [18] was the increase in the percentage of vancomycin-resistant isolates of *E. faecium* in the EU/EEA, from 10.5% in 2015 to 18.3% in 2019. The aforementioned data shows that efforts to combat the antimicrobial resistance phenomena are still in need, despite the promising improvements for controlling some bacterial species. Another point to note is that *E. coli* followed by *S. aureus* have the lion’s share of the reported bacterial species, which means that it will be essential for future therapeutics to target both types of bacteria: Gram-positive and Gram-negative.

### 1.2. The Latest Antibacterial Therapeutics Currently under Development

A number of novel approaches for fighting bacterial resistance are currently being investigated and some have even reached clinical trials. These include anti-virulence approaches which are targeted to inhibit the production or activity of virulence factors (VFs) including toxins, adhesins, quorum sensing (QS) molecules, siderophores, and immune evasion factors [20]. Another attractive strategy is microbiome-modifying therapy, which includes the manipulation and engineering of the human microbiome in order to prevent and resolve infection. This strategy has generated considerable activity in academia and industry [21]. Bacteriophages, also known as phages, have also gained traction in the last 10–15 years as a response to the emergence of multidrug-resistant pathogens. One of the characteristics of phage therapy is its specificity to single bacterial species, and usually to a subset of strains within that species [22]. Noble nanoparticles, such as gold, have also been recognized for their notable anti-biofilm efficacy [23]. Other approaches include immunotherapy, antisense RNA, drug-resistance modulation, and other approaches that have been discussed thoroughly in a recent review by Theuretzbacher U et al. (2019) [20]. In our review we have focused on the latest research on antibacterial gold nanoparticles, with a brief discussion of the different strategies used and their effectiveness in eradicating bacteria.

### 1.3. Nanomaterials as a Promising Tool for Eradicating Bacterial Resistance

As traditional drugs and antimicrobial agents fail in some cases to eradicate resistant bacteria and biofilms, the search for new tools is rising worldwide and is becoming a necessity. Several studies indicate that various types of nanomaterials (both organic and inorganic) have demonstrated promising results regarding antibacterial activity. It has also been claimed that the use of nanoparticles is one of the most promising strategies to overcome microbial drug resistance [24].

Nanomaterials are tiny particles with a range of diameter of 1–100 nm. At such a small scale the physicochemical and biological characteristics of these materials are essentially different from their bulk form [25,26]. Due to their high surface and small size effect, nanomaterials are potential candidates for applications in medical imaging, drug delivery, and disease diagnostics [25]. The size of the nanomaterials provides a large surface-area to volume ratio, which allows the binding of a large number of high affinity ligands, equipping nanoparticles with a multivalency in eradicating bacterial cells [27].

There are several mechanisms by which nanomaterials exert their antibacterial properties. These are, as illustrated in Figure 2: (i) direct contact with the bacterial cell wall; (ii) inhibiting biofilm formation; (iii) triggering of both innate and acquired host immune responses; (iv) production of reactive oxygen species (ROS); and (v) initiation of intracellular effects (e.g., interactions with DNA and/or proteins). As nanomaterials do not possess the same mechanisms of action as regular antibiotics they can be of extreme use against multidrug resistant (MDR) bacteria [28].

Of the different inorganic nanomaterials that have been researched, photothermally activated nanoparticles with absorption in the visible near-infrared (NIR) region are gaining attention. This is due to their ability to increase the local temperature in the surrounding medium upon irradiation which, consequently, deactivates different types of bacteria [29]. It is also worth mentioning that the NIR light in the so-called “biotransparent window” (750–900 nm) is considered safe for direct in vivo application and causes no damage to normal tissue (provided that irradiance limits are not breached, e.g., not exceeding 0.32 W/cm^2^ at 800 nm) [30,31].

## 2. Gold Nanoparticles (AuNPs) as Novel Antibacterial Agents

Gold as a metal is considered inert and non-toxic, although this may change when its status shifts from metallic bulk to oxidation states [32]. According to a recent literature review [29], gold nanoparticles in different dimensions and shapes are the most widely studied nanomaterials for antibacterial and anti-biofilm photothermal applications. Gold nanoparticles are now employed in many biomedical applications, including: bio-imaging, gene delivery, contrast enhancement of X-ray computed tomography, targeted drug delivery, diagnostics, plasmonic bio-sensing, colorimetric sensing, tissue engineering, photo-induced therapy, and cancer therapy [33,34].

One of the most commonly used methods to synthesize AuNPs is the chemical colloidal synthesis, which consists of a metal precursor, a reducing agent, and a stabilizer [35]. Other approaches are the biological (“green”) synthesis methods, where micro-organisms, plant extracts, or intracellular or extracellular extracts of fungi or bacteria are used in the synthesis of AuNPs [36,37]. In the last decade, gold nanoparticles in different morphologies such as spheres, rods, stars, and nanocapsules have been easily synthesized in a bottom-up approach, by adjusting the components and concentrations. Gold is multivalent; it can bind many types of ligands [38] and AuNPs have shown antibacterial ability against both Gram-positive and Gram-negative bacteria [39]. The antibacterial mechanism of action for nanoparticles (NPs) is dependent on their sizes. Smaller NPs act by forming large irreversible pores during their translocation across the bacterial cell membrane [38,39,40]. For larger NPs within the size range of 80–100 nm, although they are unable to freely translocate across the bacterial cell membrane, still several studies reported their ability to eradicate bacteria [41,42,43,44,45]. The exact antibacterial mechanism of action for larger NPs remained somewhat obscure, until in a recent study [46] researchers provided evidence for what is known as the mechano-bactericidal mechanism of non-translocating NPs. Their work demonstrated that an increase in the membrane tension of bacterial cells is caused by the adsorption of NPs leading to mechanical deformation of the membrane, and eventually cell rupture and death.

Owing to the optical and electrical properties of gold nanoparticles, they have gained increasing attention [25,40]. One particularly important feature is their localized surface plasmon resonance (LSPR), which plays an important role in many nanotechnological applications. This phenomenon occurs as the electrons on the surface of noble metal nanoparticles interact with electromagnetic radiation, generating LSPR and it is because of this that metal nanoparticles produce strong extinction and scattering spectra, beneficial in many applications [47].

Two main approaches that employ light activation in enhancing the antibacterial activity of gold nanoparticles are antibacterial photothermal therapy (APTT) and antibacterial photodynamic therapy (APDT). The unique and significant point about both of these approaches is that it is hard to induce bacterial resistance against them. In APTT, gold nanoparticles transform light into thermal energy under appropriate radiation. Gold nanorods (GNRs) and nanostars (GNSs) are utilized under this approach, for disinfecting biofilms through laser irradiation, by generating localized hyperthermia to eradicate bacteria [48,49,50]. On the other side, the APDT technique is based on irradiating photosensitizers which, in turn, generate reactive oxygen species (ROS) and thus eradicate bacteria. APDT is less efficient against Gram-negative bacteria compared with the Gram-positive forms and, in general, combining APDT with other antibacterial approaches is the best way to enhance its efficacy [47].

In our review, we categorized the antibacterial gold nanoparticles into three main groups, based upon the antibacterial approach used by researchers and as shown in Figure 3.

### 2.1. Pristine Antibacterial Gold Nanoparticles (without any External Stimuli or Attached Ligands)

AuNPs themselves are generally considered to be biologically inert and do not possess antibacterial activity [51]. In a study done to compare the minimum inhibitory concentration (MIC) value of gold nanoparticles with that of silver nanoparticles (AgNPs) against *S. aureus*, AgNPs had a MIC value of 4.86 ± 2.71 μg/mL and a minimum bactericidal concentration (MBC) of 6.25 μg/mL, while the AuNPs only began to have an inhibitory effect at a very high concentration of 197 μg/mL [52].

Despite this fact, there are some recent studies that have revealed an antibacterial action for “pristine” AuNPs in different shapes. In a study done [53] gold nanoflowers (AuNFs) and gold nanostars (AuNSTs) within a concentration of 250–500 μg/mL were shown to exhibit considerable antibacterial activity against *S. aureus*, with the AuNFs being superior. Even though the concentrations of the gold nanoparticles used were relatively high, the same particles had a good cell cytotoxic profile when tested against human dermal fibroblasts. No reduction in the viability of the cells at any concentration or any changes in cell morphology were observed. The authors attributed the antibacterial action of the AuNFs to the high-aspect ratio spikes and pillars, which induced high local stress on the bacterial cell wall causing membrane rupture.

Positively charged gold nanoclusters (AuNCs) of an average size of 2 nm were synthesized in a simple one-step, by using the cationic ligand (11-mercaptoundecyl)-N,N,N-trimethylammonium bromide (MUTAB) as a reducing and stabilizing agent [54]. These nanoclusters showed a promising antibacterial activity against MDR bacteria including both Gram-positive and Gram-negative types. Several antibacterial mechanisms of actions were identified, including the presence of a strong positive charge on the gold nanoclusters which increased their affinity to negatively charged bacteria. This caused cell membrane integrity disruption leading to cell rupture. Another element was the antibacterial effect of the capping agent used, MUTAB, and its intrinsic ability to disrupt the cell membrane of bacteria via strong electrostatic interactions. A third factor was the increase in the level of ROS generated, which increased four-fold in comparison to the control.

In an interesting study [55], researchers touched upon issues that are usually overlooked in evaluating the antibacterial activity of gold nanoparticles. These include the colloidal stability of gold nanorod (GNR) suspensions upon mixing with bacterial growth media and the possible contribution of synthesis impurities in GNR suspensions to the observed antibacterial activity. GNRs were synthesized using cetyltrimethylammonium bromide (CTAB) as a capping agent, and these nanorods were then further functionalized using different ligands to compare their stability and antibacterial action against *Staphylococcus aureus* and *Propionibacterium acnes*. Interestingly, the study found a similar antibacterial action between the supernatants from the first round of GNR centrifugation and the suspension which contained the functionalized GNRs. This stresses the importance of considering the antibacterial action and the toxicity of the chemicals used in the synthesis before asserting the source of antibacterial action. Nevertheless, after purifying the same GNRs with a second round of centrifugation, they still exhibited considerable antibacterial action, a point which makes these GNRs a potential candidate for further research in the treatment of skin follicular diseases such as acne vulgaris.

Although an antibacterial action was detected in the mentioned studies, we have seen that in some cases it may be attributed to other chemicals used in the synthesis process and not the gold nanoparticles themselves. This is an important factor to consider if we want to correctly assess the origin of the antimicrobial action. At the same time, the local stress induced by gold nanoparticles on the bacterial membranes, causing membrane rupture, is a factor that may be contributing to a considerable antibacterial effect, especially through the electrostatic interaction between bacterial membranes and gold nanoparticles due to opposite surface charges. It is also worth mentioning that there is a new area of research called mechano-bactericidal activity of nanomaterials, which promises to act against bacterial adhesion, biofilm formation, and infections with comparable effectiveness to traditional antibacterial methods [56,57,58]. The concept behind this approach is in designing antibacterial nanomaterials, with a specific nanostructure geometry capable of applying deadly mechanical forces to bacterial cells upon contact. This technique was applied in a recent study [46] for quasi-spherical and star-shaped AuNPs, where quasi-spherical nanoparticles showed a better bactericidal action due to a higher interactive affinity, causing greater membrane stretching and rupturing.

### 2.2. Antibacterial Photothermal Therapy (APTT) Based on AuNPs

Photothermal properties of particular morphologies of AuNPs like gold nanorods (GNRs), are a cornerstone in their nanomedicine and biomedical applications [59]. APTT therapy works by irradiating AuNPs with a laser source in order to produce local heat. The generated heat energy damages surrounding bacterial cells. An efficient photothermal therapy should involve an attachment of the AuNPs to the bacterial cells, so that the local heat generated will result in irreversible and permanent bacterial cell damage [60]. The APTT depends on the shape and the structure of the particles, and if they are aggregated or not [61]. Gold nanorods (AuNRs) and nanostars (AuNSTs) are two famous examples used in the APTT approach for biofilm eradication [44,46,58].

The photothermal-induced bactericidal activity of a phospholipid-decorated gold nanorod (DSPE-AuNR) suspension was explored against *Pseudomonas aeruginosa* (*P. aeruginosa*) planktonic and biofilm cultures [62]. Results showed a ~6 log cycle reduction of the bacterial viable count upon the treatment of a planktonic culture of *P. aeruginosa* with DSPE-AuNR suspension (0.25–0.03 nM) after laser irradiation, and a ~2.5–6.0 log cycle reduction of *P. aeruginosa* biofilm viable count. TEM images, as shown in Figure 4, reveal significant changes in the shape of the bacterial membrane and complete lysis upon the laser-induced treatment with DSPE-AuNR. Researchers proposed that the heat generated upon DSPE-AuNR excitation is responsible for the photo-thermolysis of bacteria in planktonic or biofilm cultures. We should add, however, that the possible contribution of the AuNRs themselves to the observed antibacterial effect should not be excluded.

In a recent study [63], using a low power laser diode, the photothermal antimicrobial activity of a chitosan-based hydrogel embedded with gold nanorods (Ch/AuNRs) was assessed against several Gram-positive and Gram-negative bacteria strains, including clinical isolates of multidrug-resistant pathogens. The results showed a promising antimicrobial activity of the Ch/AuNRs with MICs ≤ 4 μg/mL, and a very low cytotoxicity with cell viability above 80%, when tested against a murine model of macrophage cells. The authors attributed this potent antibacterial activity as a result of singlet oxygen (ROS) generation, in addition to the rupture and the autolysis of bacterial membranes due to the increase in temperature upon irradiation.

Gold nanostars (GNSs) were co-functionalized with different thiol groups and grafted as a monolayer on glass [64]. Under near-infrared irradiation (NIR) this glass proved to have an impressive antibacterial activity, capable of eliminating at least 99.99% of bacteria from both Gram-positive *S. aureus* and Gram-negative *E. coli* [64]. The irradiation power used for the photothermal activation was 0.264 W/cm^2^ (a value which is considered within the safe limits for skin exposure [31]), and at a wavelength of 808 nm which is also safe for in vivo use. Coating the GNSs with the proper thiol groups provided other benefits, including the ability to impart either high hydrophilicity or hydrophobicity to the surfaces coated with these particles. It also enhanced the shelf stability of the GNS monolayers from a few weeks to more than three months, at the same time preserving its photothermal characteristics, i.e., not compromising on antibacterial efficacy. Such coatings have a promising potential to be used in the future for coating medical devices such as catheters or prosthetic medical implants.

Photothermally activated thiol chitosan-wrapped gold nanoshells (TC-AuNSs) were developed by P. Manivasagan et al. as an antibacterial agent for the destruction of antibiotic-resistant pathogens [65]. This conjugate had several benefits as a novel antibacterial agent, including: high water solubility, biocompatibility, strong NIR absorption, and exceptional photothermal properties. At a concentration of 115 μg/mL, these nanoconjugates were capable of completely eradicating *S. aureus*, *P. aeruginosa*, and *E. coli* within 5 min of NIR laser irradiation, and no bacterial growth was detected after 48 h of laser irradiation.

What is interesting about the photothermal ablation effect is that it is less prone to bacterial resistance, something faced by many other therapeutics whether chemical or biological. Another important factor is that it can be activated on-demand and thereupon hyperthermia is only generated when particles are irradiated. This limits the side effects that may occur, although safetyremains a challenge for most of these nanosystems. However, as we will see in the following sections, some researchers have managed to tackle safety concerns by conjugating specific ligands to AuNPs, which aided in improving their targeting capabilities and in reducing adverse effects.

### 2.3. Antibacterial Photodynamic Therapy (APDT) Based on AuNPs

Photodynamic therapy (PDT) is an emerging technology for treating various diseases that require the elimination of pathological cells (e.g., tumor cells, infectious microorganisms) or the removal of unwanted tissue (e.g., atherosclerotic plaques in the arteries). It works by exciting nontoxic photosensitizers (PSs) by harmless visible light which leads to the generation of highly toxic reactive oxygen species (ROS) [66]. Light absorbed by photosensitizers generates excited triplet states which interact with molecular oxygen, creating singlet oxygen. This singlet oxygen is highly reactive and can diffuse up to 100 nm from the site of generation, causing damage to cell walls, plasma membranes, and DNA, eventually leading to the death of the microbial cells [67]. Lethal photosensitization of different microbes has been reported, including Gram-negative bacteria such as *Escherichia coli* and Gram-positive bacteria, including *Staphylococcus aureus* and MRSA [68]. However, it is worth mentioning that microorganisms appear to be much more vulnerable to lethal photosensitization than mammalian cells [69].

For PDT to be both effective and safe, it is of great importance to deliver the PSs in therapeutic concentrations to the target cells, while concomitantly ensuring they are absorbed in only minute quantities by non-targeted cells, so as to minimize unwanted side effects in healthy tissues [66]. Using AuNPs in PDT can be beneficial in two ways: first, as a drug delivery platform for PSs and second, by utilizing the surface plasmon resonance of AuNPs to enhance the PDT effect [66].

In a recent study [70], gold nanorods (AuNRs) were embedded with a photosensitizer dye crystal violet (CV) in a polyurethane (PU) matrix to fabricate an effective antimicrobial film. This film was capable of eradicating Gram-negative bacteria *E. coli* on its surface when exposed to white light. Under 3 h of light exposure this processed antimicrobial film reduced a bacterial population of 10^4^ cfu/cm^2^ to the level of 1−5 cfu/cm^2^. The film showed antimicrobial effect only when exposed to light, making it safe for use over long periods of time, and at the same time limiting the time given to the bacteria to develop resistance against it. The mechanism behind its antibacterial action was attributed to the fact that AuNRs have a broad absorption spectrum with a greater absorption coefficient than organic dyes [71], and as a result a higher capability to concentrate the energy absorbed in a localized electrical field [72]. The presence of CV dye molecules in close proximity to the AuNRs induced a strong plasmonic coupling between them, which eventually increased the amount of energy absorbed by the dye [73,74]. Consequently, this increased the amount of ROS generated by CV dye which in turn enhanced the antimicrobial efficacy of the film.

Gold nanoclusters (AuNCs) of ~2 nm in size were incorporated together with crystal violet (CV) dye into a polymer film which was activated at a low flux level of white light [75]. This treated polymer possessed a potent photobactericidal activity. More than >3.3-log reduction in viable *S. aureus* bacteria was observed after 6 h exposure of white light, while a 2.8-log reduction in the number of viable *E.coli* bacteria was observed after 24 h of white light exposure [75]. The antibacterial mechanism according to the researchers is due to the presence of AuNCs within the film matrix, which enhanced redox reactions. Once the system was activated by a low flux level of white light, an electron transfer pathway was generated from the CV dye to the AuNCs, leading to increased hydrogen peroxide formation and thus a bactericidal activity. It is worth to noting here that *E. coli* required a longer exposure time of white light to obtain a notable reduction in viable bacteria count in comparison to *S. aureus*. The authors ascribed this to the double membrane structure found in *E. coli* (a Gram-negative bacterium), whereas *S. aureus* (a Gram-positive bacterium) contains only a single membrane [76]. This double membrane of Gram-negative bacteria reduces molecular penetration, and is often responsible for an elevated resistance towards antibacterial agents [76,77].

S. Khan et al. developed a targeted antibacterial delivery photodynamic system, composed of concanavalin A (ConA, a mannose specific lectin protein) directed dextran-capped gold nanoparticles (GNPDEX-ConA), conjugated to PS methylene blue (MB) forming the nanoconjugate MB@GNPDEX-ConA [78]. This system improved the efficacy and selectivity of MB-induced killing of multidrug resistant clinical isolates, including *Escherichia coli*, *Klebsiella pneumoniae,* and *Enterobacter cloacae*. With photothermal activation, this system was capable of eradicating 97% of MDR bacteria and at the same time showed no cytotoxic effects when tested in vitro with HEK293 cells. Both dextran and ConA moieties aided in the attachment of this nanoconjugate to bacterial surface fimbriae and then bacterial surface lipopolysaccharides respectively [79,80], as shown in Figure 5 below. This in turn enhanced the system’s targeting effect. Singlet oxygen produced by the conjugated monomeric methylene blue after photoactivation was mainly responsible for the bacterial eradication. It is also worth mentioning that the presence of MB in close proximity to the gold nanoparticles increased the generation of singlet oxygen, further enhancing the antibacterial capabilities [76].

Gold nanoparticles of different shapes and sizes are able to generate reactive oxygen species (ROS) by themselves under a suitable photoactivation source [81,82]. However, in our literature review, we found that AuNPs were linked to a photosensitizer in most of the APDT systems [70,75,78]. According to the previously discussed APDT approaches, there is good scientific evidence that combining gold nanoparticles with a photosensitizer has a synergism effect and would perform better in terms of ROS generation. Yet, in spite of the promising antibacterial efficacy of the photodynamic therapy, safety and cytotoxicity should still be considered, as the same reactive oxygen species that can effectively eradicate bacterial cells might as well cause significant harm to healthy cells.

### 2.4. Antibacterial AuNPs Conjugated to Antibiotics

According to the literature [83,84], conjugating metallic nanoparticles with antibiotics tends to augment the antibacterial capabilities of the latter and, by reducing the need for high doses, mitigate side effects. The probability of bacteria developing resistance towards this antibiotic conjugated metallic nanoparticle system is also reduced [83,84]. Conjugation can be attained by both covalent and non-covalent interactions. The functionalized composites have shown greater antibacterial efficacy with lower minimum inhibitory concentration than the antibiotics alone [27].

Vancomycin, a glycopeptide antibiotic, was used as both a reducing and capping agent to gold nanoparticles of polygonal shape in a simple and fast one-pot synthesis method [85]. The conjugated vancomycin retained its antibacterial activity and the conjugate was revealed as promising: it reduced by 16-fold the amount of antibiotic needed for inhibiting certain strains of vancomycin-resistant *Enterococci* (VRE), in comparison to that of the free-form vancomycin. The system also performed well from a photothermal aspect, as in just 5 min of NIR irradiation there was a rise of approximately 15 °C (at λ = 808 nm, ∼400 mW), with a negligible level of cytotoxicity when tested in vitro. Such conjugates pave the way in the future for their use in treating VRE infections.

In another recent publication [86], gold nanoparticles were used as a drug delivery vehicle for the antibiotic colistin. Colistin is regarded as a last-resort antibiotic for multidrug resistant bacteria, and is known to have some undesirable nephrotoxic and neurotoxic side effects [87,88], which are dose-dependent. In this study, colistin was conjugated with anionic gold nanoparticles using electrostatic attraction and when tested against *E. coli* bacteria the minimum inhibitory concentration was reduced by six-fold in comparison to colistin alone. This conjugate holds great promise for delivering colistin at lower doses with improved efficacy, eventually reducing its dose-dependent side effects.

Gold nanoflowers conjugated with the antibiotic daptomycin showed promising antibacterial and antitumor effects [89]. Daptomycin micelles were used both as templates and reducing agents to produce stabilized photothermally activated daptomycin gold nanoflowers. Once irradiated by laser, an increase of about 31 °C in the temperature of the nanoconjugates was observed. This phenomenal photothermal conversion efficiency was ascribed to the special three-dimensional structure of the conjugate. These particles proved to have a promising antibacterial effect for both Gram-negative and Gram-positive bacteria. Interestingly, the same nanoflowers were also found to be biocompatible, and were efficient in inhibiting the growth of the cancerous HeLa cells in vitro. They also showed promising in vivo results, where they significantly decreased the volume of a solid tumor in Kunming mice. In this research the same conjugate proved to be multi-functional with high utility for both antibacterial and anticancer applications.

Using a one-pot reaction, the antibiotic ampicillin was conjugated with ultra-small gold nanoparticles (1.4 nm +/− 0.5 nm) which were already grafted on self-assembled rosette nanotubes (RNTs) [90], as shown in Figure 6. These RNTs were synthesized by the self-assembly of a synthetic DNA base analog, the G∧C motif. This novel conjugate showed superior antibacterial activity in comparison to ampicillin alone when tested against *S. aureus* and MRSA. Its minimum inhibitory concentration (MIC) against *S. aureus* was found to be 18% lower than ampicillin alone. It also exhibited a MIC at a concentration of 4 μg/mL against MRSA; around 10–20 times lower than reported values for ampicillin alone. Even at high concentrations of 4 μg/mL of ampicillin (70 μg/mL of AuNPs), the nanocomposite showed negligible cytotoxic effects making it an attractive option to be considered for future research and development.

Ampicillin was used in another recent study by Chavan et al. [91], where it served both as a reducing and capping agent to form ampicillin-capped gold nanoparticles (Amp-Au NPs). The synthesis process kept the β-lactam ring free to interact with bacteria. Amp-Au NPs showed promising antimicrobial activity against both ampicillin-sensitive and resistant bacteria, up to sixteen-fold and four-fold, respectively. The nanoconjugate also has been shown to be resistant against biofilm formation. According to the atomic force microscopy (AFM) and fluorescence imaging the nanoparticles accumulated on the bacterial cells, which led to the formation of pores into the bacterial membrane. These pores aided ampicillin to gain passive entry inside the bacterial cell and also sequestered the ampicillin drug from other defense barriers of the ampicillin-resistant *E. coli* bacterial strain.

### 2.5. Antibacterial AuNPs Conjugated to Phages

In a recent and pioneering study [92], researchers conjugated bacteriophages (phages) with gold nanorods (GNRs), synthesizing what they termed as “phanorods”. Chimeric phages were engineered to bind specifically to different Gram-negative organisms, including the human pathogens *Escherichia coli, Pseudomonas aeruginosa,* and *Vibrio cholerae* and the plant pathogen *Xanthomonas campestris*. The bioconjugated phanorods were able to selectively target and kill specific bacterial cells by photothermal ablation. Following irradiation by near-infrared light, gold nanorods generated heat that efficiently killed targeted bacterial cells. This system also achieved specificity in targeting *P. aeruginosa* biofilm, in which the irradiation of phanorods eradicated bacterial cells while causing minimal damage to epithelial cells. Another safety feature of this system is that the irradiation of the phanorods also destroyed the phages themselves. This prevented replication and reduced the potential risks of traditional phage therapy while enabling control over dosing. This conjugate offers an efficient, targeted, and safe antibacterial therapy that may be used as a well-controlled platform for the systematic destruction of bacterial cells.

### 2.6. Antibacterial AuNPs Conjugated to Antimicrobial Peptides (AMP)

An interesting feature of antimicrobial peptides is that they fold into an amphiphilic structures upon acting on bacterial membranes. This grants them an instantaneous permeabilization ability, against which the bacteria are unlikely to develop resistance [93]. Another fact is that polypeptides can now be manufactured on a large scale; a point which has drawn attention to cationic amphiphilic peptides as new drug candidates [94].

Novel gold nanoparticles co-functionalized with peptide moieties were prepared, and shown to possess many attractive characteristics [95]. One of the most interesting was a huge increase in the stability of the tethered peptides. They were protected from protease degradation by trypsin for several hours or a day, as compared to a few minutes in the case of free peptides [95]. In addition to that, this novel system had other beneficial characteristics including: (i) water solubility, and (ii) tethered peptides were able to fold into their functionally relevant amphiphilic α-helical structure in the presence of model membranes, thus retaining their full antimicrobial activity [95]. Enhancing the half-life of AMPs from 15 min to 24 h using gold nanoparticles will pave the way for new applications of therapeutic peptides in the biomedical field.

Hexahistidine-tagged antimicrobial peptide (HPA3PHis) was loaded onto gold nanoparticle-DNA aptamer (AuNP-Apt) forming the conjugate (AuNP-Apt-HPA3PHis) [96]. This conjugate was used as an effective therapeutic tool against a Gram-negative bacterium, *Vibrio vulnificus*, which causes fatal infections in human. When tested in vitro the intracellular infection of *V. vulnificus*-infected HeLa cells of was reduced by 90%, which in turn increased the viability of the infected cells. Furthermore, when it was tested in vivo, there was a complete inhibition of *V. vulnificus* colonization in the mouse organs that were intravenously injected with AuNP-Apt-HPA3PHis. This led to a 100% survival rate among the treated mice, whereas all the control mice died. The gold nanoparticle-DNA aptamer part AuNP-AptHis contributed to the efficient intracellular delivery of HPA3PHis into the host and, also, increased the peptide stability by protecting it from proteolysis. Another important feature of this system is that it was effective after a single administration, which makes it a cost-effective treatment and excellent in terms of administration compliance. Additional benefit of this system is that it did not exhibit any evident host toxicity, which makes it a potential candidate for pre-clinical and clinical studies in the near future.

Wong et al. [97] developed a novel gene delivery system based on antimicrobial peptide (LL37)-grafted ultra-small gold nanoparticles (AuNPs@LL37, ∼7 nm) for the topical treatment of diabetic wounds with or without bacterial infection. To this conjugate they attached pro-angiogenic vascular endothelial growth factor (VEGF) plasmids. The synthesized nanoconjugate (AuNPs@LL37/pDNAs) had many interesting therapeutic features including: a high antibacterial activity both in vitro and in vivo and an enhanced cellular and nucleus entry due to the synergistic action of the AMP with the cationic AuNPs. The presence of pro-angiogenic (VEGF) plasmids within this nanoconjugate significantly improved the gene transfection efficiency in keratinocytes resulting in promoted angiogenesis. This nanoconjugate was also capable of inhibiting bacterial infections in diabetic wounds, resulting in accelerated wound closure rates, faster re-epithelization, enhanced granulation tissue formation, and increased VEGF expression.

In an innovative approach [98], small synthetic peptide called 1018K6 conjugated to AuNPs was able to strongly maintain its antimicrobial activity by folding into a functionally relevant α-helix structure when in the presence of a membranous environment. In addition, the AuNPs enhanced the peptide bacterial killing ability, against both Gram-positive and Gram-negative bacteria. The effective concentration of 1018K6 used in the experiment was found to be in the order of <100 nM. At such sub-micromolar concentrations almost 100% of treated pathogen bacteria, such as *Listeria* and *Salmonella* genera, were eliminated. This low dose efficiency was attributed to an increase in the local concentration of the peptide surrounding each nanoparticle, that in the proximity of a bacterial membrane was more effective. With such low effective doses this will facilitate the large-scale production of these synthetic peptides in order to use them at sustainable costs.

### 2.7. Antibacterial AuNPs Conjugated to Enzymes

An antibacterial hybrid system composed of the antibiotic ampicillin (Amp) conjugated to lysozyme-capped gold nanoclusters (AUNCs) was developed [99]. This system reduced MRSA resistance to ampicillin and it remarkably increased the anti-bacterial effect of ampicillin against other non-resistant bacterial strains. When tested in vivo, on murine animal models, it eliminated systemic MRSA infection and improved the survival rate of the infected animals. Topical application of AUNC-L-Amp also eliminated MRSA infection on diabetic wounds and accelerated the healing process. According to the authors these promising antibacterial results were attributed to: (i) elevated ampicillin concentration at the site of action and an increase in its permeation, (ii) cell wall lysis caused by lysozyme, (iii) bacterial efflux pump dysfunction, and (iv) cell membrane destabilization caused by the gold ions.

Another smart photothermal nanosystem was developed, which consisted of gold nanorods (GNRs) and an adsorbed enzyme protease (protease-conjugated gold nanorods, PGs) [100]. This system was capable of causing physical damage to bacterial cells, preventing biofilm and exotoxin production, eliminating pre-existing biofilm and exotoxin, and inhibiting bacterial quorum-sensing systems. Using this PG system, the bacterial survival rate population was reduced to 3.2% and 2.1% of untreated control numbers for *E. coli* and *S. aureus,* respectively. The increase in temperature generated by the excited GNRs enhanced the protease activity in degrading the bacterial biofilm, intracellular nucleic acids, and proteins. Also, the stability of the protease was greatly enhanced after the immobilization onto GNRs surfaces, due to protection from bacterial inactivation. The unique element of this novel nanosystem is that it addressed the issue of persistence of bacterial residues that perpetuate chronic illness in patients even after live bacteria have been eliminated.

### 2.8. Novel Antibacterial Vaccines Based on AuNPs

A novel synthetic nanogold-based vaccine system against entero-hemorrhagic *Escherichia coli* (EHEC) has been recently developed at the university of Texas [101]. Two chosen EHEC-specific immunogenic antigens, namely LomW and EscC, were linked covalently to AuNPs to form stable formulations of AuNP-LomW and AuNP-EscC. These conjugates were used to immunize mice before being challenged with a specific strain of EHEC bacteria. Higher levels of Immunoglobulin G IgG titers in serum and secretory Immunoglobulin A IgA titers in the feces of the immunized mice were measured after around 35 days of subcutaneous administration. The elevated level of IgG titers correlated with a significant decrease in EHEC intestinal colonization after three days post inoculation. Additionally, serum from antigen-coated AuNP-immunized mice has been shown to reduce the adherence of human intestinal epithelial cells for EHEC, as well as for two other *E. coli* pathotypes, when tested in vitro. The serum also showed antigen-specific bactericidal properties, enhancing the classical complement pathway. The success of this synthetic nanogold vaccine against *E. coli* will open the door for the development of more synthetic nanogold-based vaccines in the future against other harmful pathogens.

### 2.9. Theranostic Antibacterial Systems Based on AuNPs

H.Wang et al. synthesized in a recent study [102] a novel, intelligent, and safe theranostic system based on a bacteria-induced gold nanoparticle (GNP) aggregation, offering both high levels of efficiency for bacterial surface-enhanced Raman scattering (SERS) imaging and for antibacterial photothermal therapy. By employing the bioorthogonal cycloaddition technique, tetrazine-modified gold nanoparticles (GNP-Tz) orthogonally reacted with a trans-cyclooctene derivative of vancomycin (Van-TCO) in situ via instantaneous cycloaddition to form aggregated GNPs on the bacterial surface, as shown in Figure 7. A plasmon coupling effect was generated between adjacent GNPs which induced a strong electromagnetic field and high NIR absorption. Due to this, an effective surface-enhanced Raman scattering (SERS) imaging and photothermal ablation of the bacterial pathogens was achieved. The unique thing about this system is that in the absence of bacterial cells, GNPs were dispersed and showed very low levels of photothermal activity, which minimized side effects on the surrounding healthy tissues while maintaining a targeted bactericidal effect.

### 2.10. Antibacterial Pre-Treated Macrophage-Membrane-Coated Gold-Silver Nanoparticles

An innovative targeted macrophage-membrane-coated nanosystem was developed by C. Wang et al. [103], by which *S.aureus* pretreated macrophage membranes where attached to gold–silver nanocages (GSNCs) forming the nanoconjugate Sa-M- GSNC. This system offered many advantages: (i) the ability to adhere specifically to bacterial cells for targeted therapy; (ii) the application of the photothermal ablation effect, which resulted in significantly reduced bacterial counts both in vitro and in vivo; (iii) a unique structure (of hollow interiors and porous walls) of the GSNC, where antibacterial drugs can be loaded, and released with an on-demand control under NIR light; and (iv) an excellent biocompatibility profile and prolonged blood circulation time when tested in vivo on mice. This research will open further door and insights for future antibacterial therapies based on the same concept.

### 2.11. Gold Nanosystems Targeting Bacterial Biofilms:

As discussed earlier in the review, biofilms are structural and functional bacterial communities, where bacterial cells are encapsulated within a hydrated extracellular polymeric substance (EPS), which can attach to both biotic and abiotic surfaces [104]. Biofilms can shield bacteria from antibiotics, the host immune system, and harsh external physical or chemical environments [105]. The concentration of antibiotics for eradicating biofilms can range from 100 to 1000 times that of the minimum inhibitory concentration (MIC) needed eliminate free bacteria [17,106]. All of these facts combined urge researchers to search for more effective solutions to deal with the health challenges imposed by biofilms.

In a recently published study [107], a novel multifunctional gold nanosystem, composed of deoxyribonuclease (DNase)-functionalized gold nanoclusters (AuNCs) was formulated by Y. Xie et al. This nanosystem was capable of eliminating Gram-positive and Gram-negative bacteria, and also dispersing the surrounding biofilms. Three antibacterial mechanisms were employed synergistically, in which the DNase’s role was breaking down the extracellular polymeric substance matrix. This in turn, exposed the bacteria to photothermal therapy (PTT) and photodynamic therapy (PDT) by DNase-AuNCs under 808 nm laser irradiation. As a result, the treated biofilms were removed with a dispersion rate of up to 80% and ∼90% of the shielded bacteria were eradicated, with a short treatment time (10 min of incubation and 10 min of illumination). This nanosystem showed also an outstanding therapeutic effect in treating bacterial biofilm-coated orthodontic devices, which makes it of great potential in future biomedical applications.

Another innovative anti-biofilm approach that employs gold nanoparticles is the laser-induced vapor nanobubbles (VNBs). Using this state-of-the-art technique [108], biofilms of both Gram-negative (*Burkholderia multivorans*, *Pseudomonas aeruginosa*) and Gram-positive (*Staphylococcus aureu*s) bacteria were loaded with cationic 70 nm gold nanoparticles, which gradually penetrated through sessile bacterial cells. Subsequent laser illumination resulted in a notable increase in temperature, causing the water surrounding AuNPs to quickly evaporate in the form of water vapor nanobubbles [109,110]. VNB formation inside the biofilms disturbed the biofilm integrity and increased the space between sessile cells, allowing antibiotics to reach the target cells more easily, even deep within the dense cell clusters. In all types of biofilms tested, tobramycin efficacy increased up to 1–3 orders of magnitude depending on the organism and treatment conditions. This makes laser-induced VNB a promising strategy to eradicate biofilms effectively, by improving antibiotic diffusion.

### 2.12. Antibacterial AuNPs Conjugated to Proteins

Sun et al. [111] showed that co-functionalizing AuNPs with both bovine serum albumin (BSA) and 4,6-diamino-2-pyrimidinethiol (DAPT) can generate conjugates (Au_DAPT_BSA) with enhanced antimicrobial efficacies, including decreased minimal inhibitory concentrations against Gram-negative bacteria and extended antibacterial spectra against Gram-positive bacteria compared with DAPT-capped Au NPs (Au_DAPT). This novel conjugate (Au_DAPT_BSA) did not induce drug resistance and could significantly lessen the number of bacteria in the biofilms formed by *P. aeruginosa* and *S. aureus*. It also enhanced healing when it was tested in vivo on mice with subcutaneous abscesses caused by clinically MDR *E. coli* or *S. aureus* without inducing detectable toxicity to the mammalian cells in animals.

#### 2.12.1. Antibacterial AuNPs conjugated to Aminosacharrides

D-glucosamine (GluN)-modified gold nanoparticles (Au_GluN) showed the best antimicrobial activities among other AuNP-based multivalent aminosaccharides in a study reported by X. Yang et al. [112]. The AuNP-based multivalent aminosaccharides could effectively and selectively inhibit the growth of Gram-positive bacteria (including drug-resistant types like MRSA). The remarkable efficiency was due to the similarity between the peptidoglycan layer of the bacterial cell wall and the tethered aminosaccharide. The conjugate was capable of changing the permeability of the bacterial cell membrane and disrupting the cell wall thereby leading to bacterial death. Results were further confirmed by the same research group in a recent in vivo study [113] where the same conjugate lowered the bacterial viability in a mature biofilm and showed high efficiency in healing a superbug-infected wound in mice.

#### 2.12.2. Microbiota Friendly Antibacterial AuNPs Targeted Therapy

A new antibacterial targeted therapeutic composed of 4,6-diamino-2-pyrimidinethiol DAPT-coated Au (D-Au NPs) was developed by Li et al. [114]. In comparison to conventional antibiotics, which disrupt intestinal microflora, this orally delivered nanoconjugate was capable of curing infections induced by *E. coli* in mice gut without compromising the integrity of intestinal microflora. D-Au NPs showed no liver or kidney toxicity when tested after 28 days, and it was harmless to intestinal epithelial cells. The mechanism of action behind this nanoconjugate was attributed to the D-Au NPs acting upon the cell membrane of *E. coli* and causing it to rupture. The specificity of such nanoconjugates will pave the way for developing new antibacterial therapeutics, where the individual’s digestive system health is not compromised for the curing benefits of antibiotics.

We summarized all the previously discussed studies of the different antibacterial AuNPs approaches in Table 2. 

## 3. Toxicity of Gold Nanoparticles

Since we are reviewing the therapeutic potential of gold nanoparticles (AuNPs), safety is a major factor that needs to be considered. The general prevailing opinion that AuNPs are non-toxic is now subject to discussion. The toxicity caused by AuNPs whether in vitro or in vivo seems to be multifaceted and hard to predict. Some in vitro researches have shown that AuNPs, once incorporated inside cells, generate endogenous reactive oxygen species (ROS), which then leads to further oxidative stress-related cytotoxicity such as DNA damage, cell death, and eventually cell-cycle arrest [115]. In an in vitro viability assay, AuNPs within the size range of 15–20 nm decreased the viability of human cells from 100% at a concentration of 0.1 ppm to less than 40% at a concentration of 10 ppm, illustrating a significant dose-related toxicity for AuNPs [116]. Although the scope of our review is not mainly focused on assessing the toxicity of AuNPs, we will spotlight on several studies both in vitro and in vivo discussing the outcomes and factors which may affect the safety and pharmacokinetics of AuNPs. Qiyue Xia et al. [117] discussed in a comprehensive and detailed review the elements affecting the pharmacokinetics, biodistribution, and toxicity of AuNPs in drug delivery.

### 3.1. Effect of Size and Shape on the Toxicity of AuNPs (In Vitro)

In a study [118] to assess whether there was a relation between the size and cytotoxicity of AuNPs, researchers investigated the effects of citrate-stabilized AuNPs in vitro on Balb/3T3 mouse fibroblasts. Results obtained, after exposing the cells for 72 h to AuNPs of 5 and 15 nm in size, showed cytotoxic effects only for the 5 nm particles at a concentration of ≥50 μM. A size-dependent cytotoxic effect of the AuNPs was reported, although the exact mechanism behind it still needs further investigation.

On the other hand, T. Mironava et al. [119] found that 45 nm particles were more toxic than 13 nm ones. This might be due to the higher damaging effect of the 45 nm AuNPs on vacuoles and subsequently to the greater release of these particles into the cytoplasm, which resulted in disruption of the normal cell function. So although there is some sort of relation between the different sizes and cytotoxicity of AuNPs, drawing precise conclusions from literature is still difficult due to the variation in results obtained [120].

The effects of shape (spheres and stars), size (14 nm and ∼50 nm), and capping agent (11-mercaptoundecanoic acid (MUA) and sodium citrate) of AuNPs on the cytotoxicity of cells have been compared in a systematic multi-parametric comparative study [121]. They found that toxicity was greater for stars when compared with sphere-shaped AuNPs, and that citrate coating was more toxic than MUA. To compare the effect of size, the researchers evaluated the differently sized AuNPs based upon the number of AuNPs/volume unit instead of the more commonly used Au atom concentration (in mass or mol/volume unit); and accordingly a higher degree of cytotoxicity was noted for the larger 50 nm AuNPs.

### 3.2. Effect of Surface Functionalization on the Toxicity of AuNPs (In Vitro)

One of the essential characteristics of noble nanoparticles is the ease of modifying their surface. The aim of surface functionalization is to improve their intrinsic properties such as absorption profiles, stability, targeting capabilities, and therapeutic outcomes. Furthermore, surface modification can aid in overcoming challenges arising from the in vivo environment such as adsorption of cells, thiols, antibodies, and proteins; detection by reticuloendothelial system (RES); and cell uptake processes [122].

In a study to compare the effect between two different surface modifiers on AuNPs, Muoth et al. [123] found that smaller nanoparticles of 3–4 nm or sodium carboxylate-modified AuNPs had an increased uptake compared to larger ones (13–14 nm) or polyethylene glycol (PEG) coated AuNPs (PEGylated AuNPs). Similarly, in another study [124] the uptake of citrate-capped AuNPs (CitAuNPs) was compared to PEGylated ones (COOH-PEGAuNPs) and the uptake was found to be higher for the citrated ones. It is worth noting that the cell lines used to conduct these studies were not the same and the methods used to evaluate the uptake also differed [123,124]. AuNPs conjugated with both peptide and polyethylene glycol (PEG) had enhanced uptake in HeLa cells in comparison to AuNPs conjugated only with PEG [125]. Charge present on the surface of gold nanoparticles also greatly influences cellular toxicity. Positively charged nanoparticles are more easily transported into cells, due to the electrostatic interaction with the negatively charged cell membrane, and this results in the breakage of cell membranes. On the contrary, the anionic surface groups’ functionalized gold nanoparticles were found to be more safe [126,127].

### 3.3. The In Vivo Toxicity and Biodistribution of AuNPs

Biodistribution is one of the most important aspects associated with nanoparticle-enabled drug delivery, determining efficacy and toxicity [128]. A. L. Bailly et al. [129] evaluated in a recent study the in vivo toxicity of laser-ablated dextran-coated AuNPs (AuNPd). They showed that these AuNPd were rapidly eliminated from the blood circulation of mice and accumulated preferentially in the liver and spleen, without causing kidney or liver toxicity. Despite certain residual accumulation in tissues, there were no signs of histological damage or inflammation in tissues and the interleukin (IL-6) level confirmed the absence of any chronic inflammation. The safety of AuNPd was also assessed by the healthy behavior of mice and the absence of acute and chronic toxicities in liver, spleen, and kidneys. The promising safety profile of this study may shift the focus of the commonly adopted chemical synthesis methods of AuNPs towards using physical techniques where less impurities are utilized and the therapeutic outcomes can be attributed to the AuNPs themselves and not the chemical impurities. The in vivo safety of AuNPs has been recently evaluated by R. Han et al. [130], where they developed a twophoton photodynamic therapy (TP-PD) system based on the nanoconjugate dihydrolipoic acid-coated gold nanocluster (AuNC@DHLA). This conjugate had an extremely high two photon (TP) optical properties of ∼10^6^ Goeppert-Mayer (GM), with strong ROS generation ability. When tested in vivo it showed high efficiency against a hepatocellular carcinoma xenograft tumor mouse model, with negligible toxicity and an excellent biocompatibility profile.

### 3.4. Effect of Surface Functionalization on the Biodistribution and the Toxicity of AuNPs (In Vivo)

Takeuchi et al. [131] reported a comparison in the biodistribution profile of PEGylated-AuNPs versus non-PEGylated AuNPs with diameters of 20–30 nm and 50 nm. They found that at 48 h after intravenous administration, accumulation in the liver and spleen was notably decreased by PEGylation, and the gold amounts of PEGylated gold nanoparticles with diameters of 20–30 nm and 50-nm in the brain were 3.6 times and 2.7 times higher than those of bare gold nanoparticles, respectively. In a related study, it has been reported by Velasco-Aguirreet et al. [132] that Au nanorods with both PEG and angiopep-2 can accumulate in the brain, in contrast to those only functionalized with PEG. The study provided evidence that angiopep-2 can cross the blood brain barrier (BBB) and improve the delivery of Au nanorods to brain parenchyma. AuNPs with three different surface coatings consisting of neutral (PEG), anionic lipoic acid (LA), or cationic branched polyethyleneimine (BPEI), and of two different sizes of 40 or 80 nm, were evaluated for their toxicity and biodistribution in an isolated, perfused ex vivo porcine skin preparation [133]. Toxicological effects were not detected, and the study showed that arterially infused 40 or 80 nm AuNPs of the three different surface coatings with or without defined protein coronas were distributed to perfused skin without adverse vascular effects (e.g., changes in glucose utilization, vascular resistance), which supports the use of AuNPs for intravenous nanomedicine applications. Another important finding was that cationic branched polyethyleneimine BPEI-AuNPs of both sizes had preferential tissue accumulation compared to the other coatings, even after exposure and perfusion in a complex protein-containing medium. This finding paralleled previous in vitro cell culture studies using the same AuNPs [134,135,136,137] in human cells, showing that BPEI-AuNPs had also the greatest uptake. This kind of studies will open the door for constructing physiologically based pharmacokinetic (PBPK) models capable of employing available in vitro data and to validate them in an in vivo context.

### 3.5. Effect of Shape and Size on the Biodistribution and the Toxicity of AuNPs (In Vivo)

Using an adult zebrafish model for an in vivo study [138] Sangabathuni et al. compared three different shapes of PEGylated and mannose-AuNPs (sphere, rod, and star) after being intraperitoneally injected into the fish. Very low toxicity was detected, indicating the potential use of these nanoparticles for drug delivery and imaging studies. PEGylated-AuNPs had less sequestration than the mannose-AuNPs. Mannose-AuNPs were present in the digestive system, heart, and swim bladder, but not in the muscles, brain, or the eyes. After 24 h and 48 h, the shape-dependent accumulation of nanoparticles appeared. Initially, rods accumulated in a higher number and were cleared after 48 h, whereas, the star-shaped particles accumulated in a steady state and were sequestered for a longer time when compared to the spheres. Talamini et al. [128] analyzed the biodistribution profile of AuNPs with different shapes and sizes. Their biodistribution study revealed that the same amounts of spherical and star-like AuNPs accumulated in the liver, however in different locations. Additionally, only star-like AuNPs were found to accumulate in the lung. The accumulation of larger AuNPs (50 nm) was fast in the liver and spleen, and their increase with time was significant. On the other hand, smaller AuNPs (10 nm) showed a progressive increase of levels in tested animals. In a review by Schmid G et al. [139] ultra-small AuNPs (usAuNPs), with a size smaller that smaller than 2 nm, exhibited remarkably distinct biodistribution and enhanced circulation times compared to larger AuNP. In contrast to larger particles, which accumulate rather quickly in the liver, usAuNPs tend to distribute over all other organs as well.

As we have seen, the safety/toxicity of AuNPs is a multifactorial process. The concentration, size, shape, and surface charge/functionalization of the particle, as well as other factors, all play a major role in determining how toxic a specific conjugate is and where it is distributed throughout the body. Available data from the literature is conflicting in many cases, and this is can be attributed to different settings of experiments and to the absence of standardization protocols when relatively similar nanoparticles are studied. Accordingly, it is still difficult to conclude or agree on specific points concerning the safety of AuNPs. However, if we look at the glass as being half full, in several in vivo studies the results are promising in terms of effectiveness and biocompatibility. What we hope for in the future is a greater collaboration between different research groups, and thus by standardizing methods and experiments more robust and dependable results will be obtained.

### 3.6. Gold Nanoparticles in Clinical Trials

Although the safety and toxicity of the AuNPs is still a topic that needs to be further addressed, there are some promising studies that feature both an excellent safety profile along with potential therapeutic benefits. To the best of our knowledge no clinical trials have been done on antibacterial gold nanoparticles to date. However, an interesting clinical trial [140] has been done at Mount Sinai hospital in New York, in which photothermally activated gold-silica nanoshells (GSNs) were IV infused to 16 patients diagnosed with low- or intermediate-risk localized prostate cancer. The infusion was accompanied by magnetic resonance ultrasound fusion imaging and high precision laser ablation to focally ablate low–intermediate grade tumors within the prostate. Results were outstanding and the therapy proved to be successful in 94% (15/16) of patients. This treatment protocol appears to be suitable and safe in men with low- or intermediate-risk localized prostate cancer without causing serious complications or detrimental changes in genitourinary function. Although this clinical study is targeted towards cancer treatment, its outstanding safety and efficacy will hopefully pave the way for future antibacterial nanogold clinical trials.

## 4. Conclusion and Future Perspectives

After discussing the most recent research and progress in the antibacterial gold-based nanoparticles field, we go back to the question we first proposed. Will gold nanoparticles be the next magic bullet for combating the superbugs endangering our world? In order to conclude, the answer will be a little longer than a simple yes or no.

As we have seen, gold nanoparticles are versatile in the way that they interact with bacteria and overcome its resistance mechanisms. From mechanical stress, caused by gold nanoparticles acting upon the bacterial cell walls, to the hyperthermia and ROS generated by photo-activation, researchers have observed that in many cases these resistant bacteria stand defenseless against these new therapeutic strategies. The mechanisms of action behind the different antibacterial gold nanoparticles still need to be further validated and understood, but the fact is that they are offering new hope in a time where bacterial resistance is still rising against almost any other available treatment option.

It is obvious that there are many AuNPs antibacterial-based approaches and that most of them are promising, but we think much more research should be focused on evaluating the safety of these systems. Long-term in vivo and in vitro biosafety and experimental observations are crucial in transferring these innovative therapeutics into clinical practice and in guiding their development. Another area where we think improvement is needed is the standardization of experiments, in order to obtain comparable studies.

The era of nanomedicine has just started. We hope in the near future to have safe and effective nanogold-based therapeutics, where gold’s capabilities are fully exploited to eradicate the threatening superbugs in the world once and forever.

## Figures and Tables

**Figure 1 nanomaterials-11-00312-f001:**
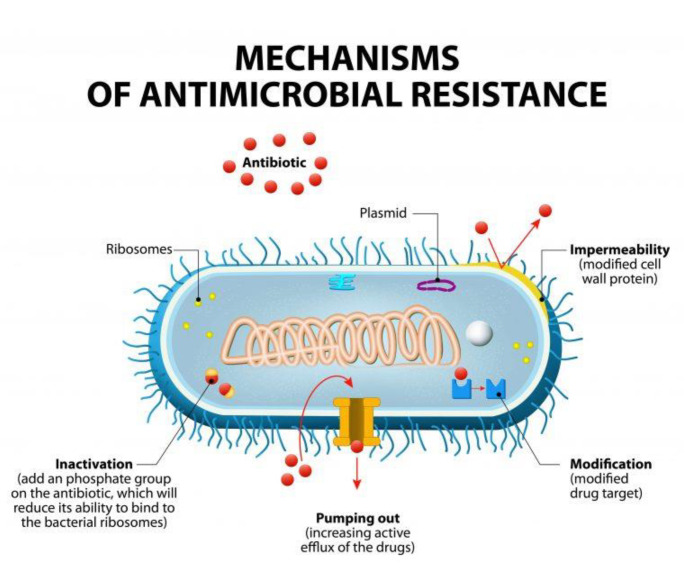
Mechanisms of bacterial resistance to antibiotics (from Designua/Shutterstock.com).

**Figure 2 nanomaterials-11-00312-f002:**
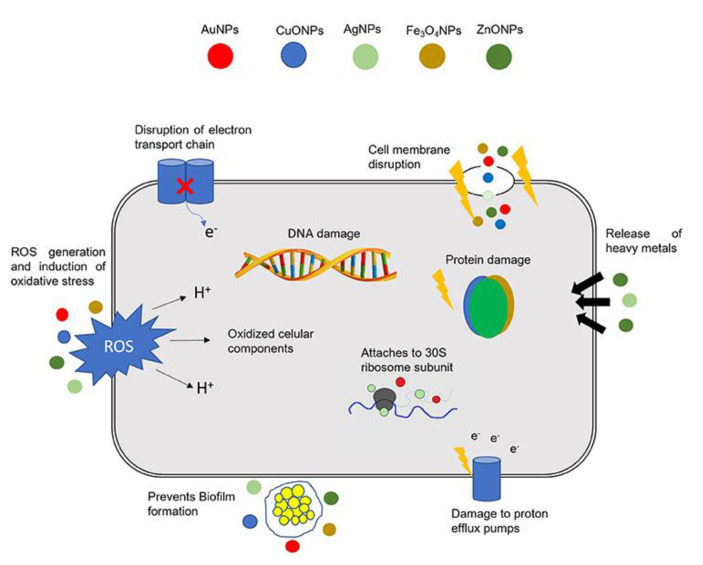
Different mechanisms of action of nanoparticles (NPs) in bacterial cells. The combination of a multitude of cellular effects in a single nanomaterial may have a tremendous impact in fighting multidrug resistant (MDR) bacteria. DNA, deoxyribonucleic acid; ROS, reactive oxygen species; AuNPs, gold NPs; CuONPs, Copper oxide NPs; AgNPs, silver NPs; Fe3O_4_NPs, iron oxide NPs; ZnONPs, zinc oxide NPs. Reproduced from [28], with permission from Front. Microbiol., 2018.

**Figure 3 nanomaterials-11-00312-f003:**
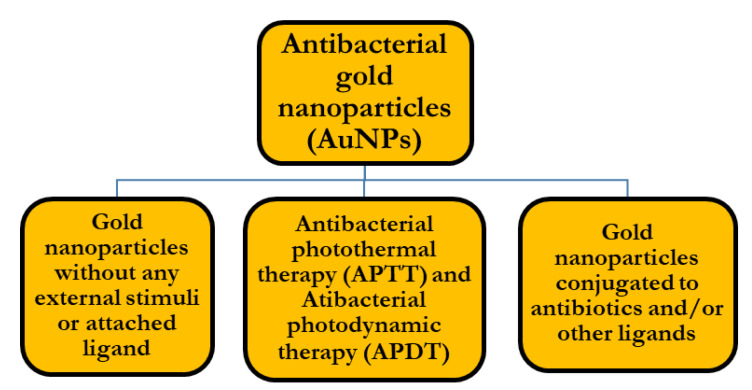
Classification of approaches used in antibacterial gold nanoparticles.

**Figure 4 nanomaterials-11-00312-f004:**
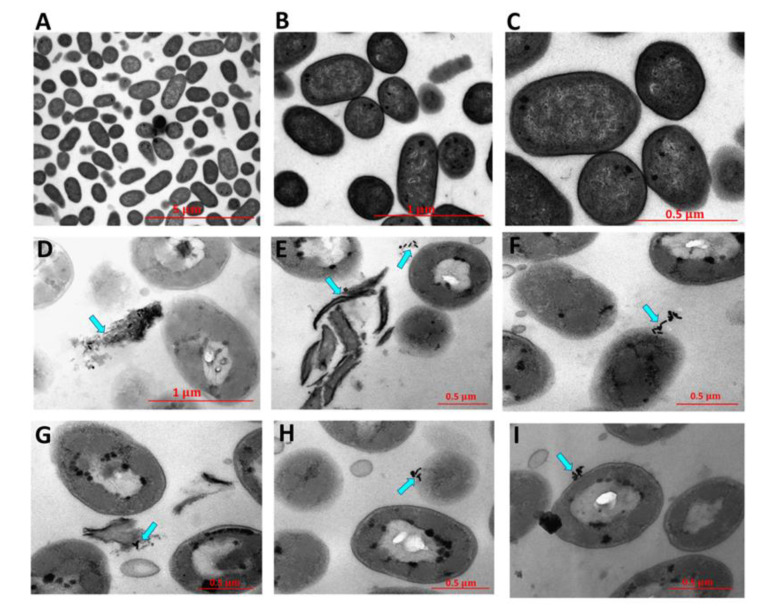
TEM images of untreated *Pseudomonas aeruginosa* (**A**–**C**) and after treatment with DSPE-AuNR and excitation with a continuous (CW) laser beam (**D**–**I**). Photothermal therapy resulted in significant changes in the morphology of the bacteria and lysis of bacterial cells. Reproduced from [62], with permission from MDPI, 2019.

**Figure 5 nanomaterials-11-00312-f005:**
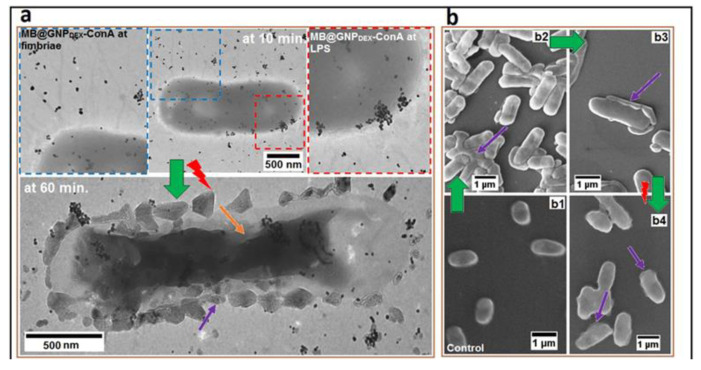
(**a**) TEM micrographs of localized MB@GNPDEX-ConA on *K.Pneumoniae*-12 bacterial surface and treated (for 100 s = 142.9 J cm^−2^) bacterial cell (after 60 min). The micrograph shows the cytological mass with aggregated nanoconjugates (yellow arrow) and cell surface perturbation (violet arrow). (**b**) The morphological perturbations shown by SEM micrographs. (**b**,**b1**) The intact and uniform morphology of control cells and (**b**,**b2**,**b3**) bacterial aggregation due to concanavalin A (ConA)-mediated attachment of nanoconjugates (violet arrow). (**b**,**b4**) The micrograph shows cell membrane destruction (violet arrow) after photosensitization. Reproduced from [78], with permission from Elsevier, 2017.

**Figure 6 nanomaterials-11-00312-f006:**
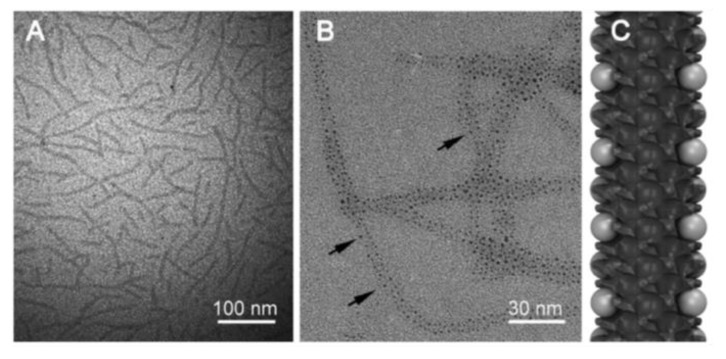
TEM images of (**A**) PEG-RNTs, (**B**) AuNP/PEG-RNT nanocomposite, and (**C**) model of the nanocomposite. Black arrows in (**B**) point at the AuNPs grown on the PEG-RNT surface. Abbreviations: RNT, rosette nanotube; PEG, polyethylene glycol; TEM, transmission electron microscopy; AuNP, gold nanoparticle. Reproduced from [90], with permission from Dove Medical Press, 2019.

**Figure 7 nanomaterials-11-00312-f007:**
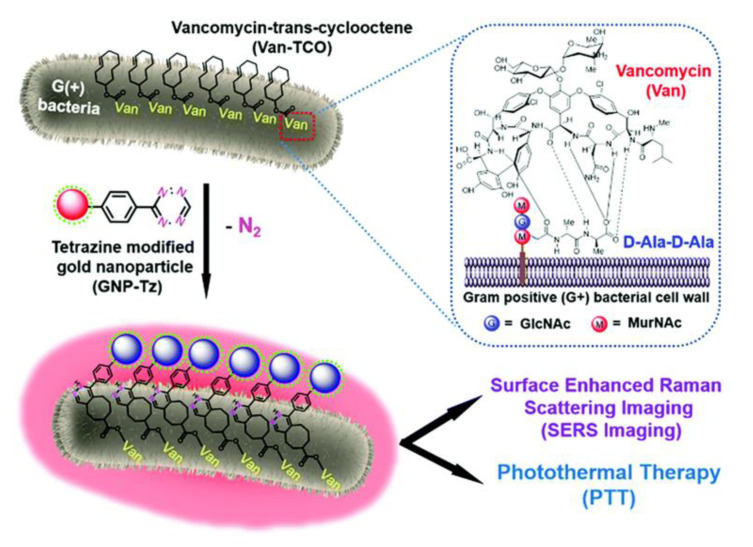
Scheme diagram of bacteria-induced gold nanoparticle aggregation for SERS imaging and enhanced photothermal ablation of Gram-positive bacteria. Firstly, vancomycin-trans-cyclooctene (Van-TCO) attached to the Gram-positive bacterial surface through five hydrogen bonds. Next, tetrazine-modified gold nanoparticle (GNP-Tz) conjugated with Van-TCO and in situ aggregated onto the bacterial cell wall via bioorthogonal cycloaddition, providing high efficiency for bacterial SERS imaging and photothermal antimicrobial therapy. Reproduced from [102], with permission from the Royal Society of Chemistry, 2019.

**Table 1 nanomaterials-11-00312-t001:** Percentage of reported bacterial species according to the EARS-Net report in 2019 [18].

Bacterial Species	Reported Percentage
*E. coli*	44.2%
*S. aureus*	20.6%
*K. pneumonia*	11.3%
*E. faecalis*	6.8%
*P. aeruginosa*	5.6%
*S. pneumoniae*	5.3%
*E. faecium*	4.5%
*Acinetobacter species*	1.7%

**Table 2 nanomaterials-11-00312-t002:** Summary of all the antibacterial AuNPs approaches discussed in previous section.

Antibacterial Approach Used	Size, Morphology, and Conjugated Entities (If Present)	Type of Bacteria	Efficacy	Ref.
Pristine antibacterial AuNPs	(1) Gold nanostars (AuNSTs) 26.0 ± 2.6 nm, gold nanoflowers (AuNFs) 40.6 ± 2.2 nm.	*S. aureus*	AuNSTs and AuNFs caused a decrease in the exponential growth rate of bacteria ~59% and 76%, respectively, upon the addition of 500 μg/mL.	[53]
	(2) Positively charged gold nanoclusters (AuNCs) ~2 nm.	*Bacillus subtilis, Enterococcus faecalis*, *Streptococcus pneumoniae,* vancomycin-resistant *enterococcus* (VRE)*, E. coli, P. aeruginosa,* and *Miconia albican*	MIC value against all tested bacteria did not exceed 4 μg/mL.	[54]
	(3) Gold nanorods (GNRs) with a length and width of ∼49.5 nm and ∼12 nm, respectively.	*P. acnes* and *S. aureus*	(Purified) GNRs had a higher MIC compared to unpurified ones, which shows that the impurities have a major contribution in the antibacterial action.	[55]
	(4) Gold nanostars (AuNSTs) had an average diameter of 103.9 ± 11.9 nm and quasi-spherical AuNPs had an average diameter of 100 ± 20 nm.	*P. aeruginosa* and *S. aureus*	AuNPs had a better antibacterial activity than AuNSTs, and the hydrophilic AuNPs had a better antibacterial efficacy in comparison to the hydrophobic ones, where a 100% inactivation of bacteria was achieved at a concentration of ≥50 µg Au mL^−1^.	[46]
APTT (antibacterial photothermal therapy)	(1) Phospholipid-decorated gold nanorods (DSPE-AuNR) with an average width and length of 49.8 ± 2.6 nm and 11.8 ± 1.8 nm, respectively.	*P. aeruginosa*	0.25–0.03 nM followed by laser irradiation resulted in ~6 log cycle reduction of the planktonic bacteria, and ~2.5–6.0 log cycle of viable biofilm count.	[62]
	(2) Chitosan-based hydrogel embedded with gold nanorods (Ch/AuNRs) with an average length and width of 49.9 ± 2.95 nm and 10.6 ± 0.78 nm, respectively.	*S. epidemidis*, *S. aureus Acinetobacter baumanni, E. coli,* and *P. aeruginosa*	MIC value against all tested bacteria did not exceed 4 μg/mL.	[63]
	(3) Gold nanostars (GNSs) co-functionalized with different thiol groups grafted on glass.	*E. coli* and *S. aureus*	At least 99.99% of the bacterial strains were eradicated after photothermal activation.	[64]
	(4) Photothermally activated thiol chitosan-wrapped gold nanoshells (TC-AuNSs) of nearly spherical shape with an average diameter of 185 ± 19 nm.	*S. aureus*, *E. coli,* and *P. aeruginosa*	TC-AuNSs (115 μg/mL) were capable of completely destroying *S. aureus*, *P. aeruginosa*, and *E. coli* within 5 min of NIR laser irradiation, and no bacterial growth was detected after 48 h of laser irradiation.	[65]
APDT (antibacterial photodynamic therapy	(1) Gold nanorods (AuNRs) of 20 to 30 nm in length and 7 to 14 nm in diameter embedded with a crystal violet dye in polyurethane (PU) matrix film.	*E. coli*	Reduction of bacterial level upon contact with film from 10^4^ cfu/cm^2^ to the level of 1–5 cfu/cm^2^ in 3 h of light exposure.	[70]
	(2) Gold nanoclusters of ~2 nm size incorporated with crystal violet dye into a polymer film.	*S. aureus* and *E. coli*	>3.3-log reduction in viable *S. aureus* bacteria after 6 h exposure of white light, and a 2.8-log reduction in viable *E. coli* bacteria after 24 h of white light exposure.	[75]
	(3) Concanavalin A directed dextran-capped gold nanoparticles (GNPDEX-ConA) ~23 nm size with nearly hexagonal symmetry, conjugated to methylene blue.	*E. coli*, *Klebsiella pneumoniae,* and *Enterobacter cloaca*	With photothermal activation, the nanoconjugate was capable of eradicating 97% of MDR bacteria.	[78]
Gold nanoparticles conjugated to antibiotics	(1) Vancomycin-immobilized gold nanoparticles (Au@Van NPs) with polygonal shapes.	Vancomycin-resistant *Enterococci* (VRE)	The MIC50 of vancomycin loaded on Au@Van NPs using photothermal approach was 2 μg/mL, which was much lower than that of free-form vancomycin.	[85]
	(2) Colistin conjugated to 5 nm diameter gold nanoparticles.	*E. coli*	With the conjugate of anionic gold nanoparticle and colistin, the minimum inhibitory concentration of *E. coli* was reduced six-fold compared to antibiotic alone.	[86]
	(3) Daptomycin-tethered gold nanoflowers with a diameter of either ~30 or 80 nm (depending on the molar ratio of chloroauric acid to daptomycin).	*E. coli* and *S. aureus*	The antibacterial inhibition rate of the 80 nm sized nanoconjugate was 64% for *S. aureus* and 52% for *E. coli*.	[89]
	(4) Ampicillin tethered on ultra-small gold nanoparticles (1.4 nm +/− 0.5 nm) which were already grafted on self-assembled rosette nanotubes (RNTs).	*S. aureus* and MRSA	MIC against *S. aureus* was 18% lower than ampicillin alone. MIC at a concentration of 4 μg/mL against MRSA was around 10–20 times lower than reported values for ampicillin alone.	[90]
	(5) Ampicillin conjugated to nearly spherical AuNPs in size range between 25 and 50 nm.	*E. coli*, *S. aureus, B. subtilis,* and *Flavobacterium devorans*	The efficacy of Amp-Au NPs increased against both ampicillin-sensitive and ampicillin-resistance bacteria up to sixteen-fold and four-fold, respectively in comparison to ampicillin alone.	[91]
Gold nanoparticles conjugated to phages	Bacteriophages conjugated to gold nanorods of an average length of 53.2 nm and average width of 13.7 nm.	*E. coli*, *P. aeruginosa*, *Vibrio cholerae,* and the plant pathogen *Xanthomonas campestris*	Roughly 50% of *E.coli* bacteria were killed after 3 min, ∼80% of bacteria were killed after 6 min, and no viable bacteria remained after 10 min of photo-activation. Similar results were observed for the other host bacterial cells.	[92]
Gold nanoparticles conjugated to antimicrobial peptides (AMP)	(1) Gold nanoparticles with a mean diameter of around 5–7 nm were covered with five different types of cationic antimicrobial peptides (AMPs).	*E.coli*, *S.aureus, Bacilus subtilis*, and *Micrococcus luteus*	The goal of the study was to enhance the stability of the AMPs while retaining their antibacterial activity, and this was achieved by conjugating these AMPs to gold nanoparticles.	[95]
	(2) Hexahistidine-tagged antimicrobial peptide (HPA3PHis) loaded onto gold nanoparticle-DNA aptamer (AuNP-Apt) forming the conjugate (AuNP-Apt-HPA3Phis) with an average diameter of 874.7 ± 232.8 nm.	*Vibrio vulnificus*	When tested in vitro the intracellular infection of *V. vulnificus*-infected HeLa cells was reduced by 90%, and when tested in vivo there was a 100% survival rate among the treated mice, whereas all the control mice died.	[96]
	(3) Antimicrobial peptide (LL37) grafted on ultra-small gold nanoparticles ∼7 nm (AuNPs@LL37) and conjugated with pro-angiogenic (VEGF) plasmids.	MRSA	Almost all the MRSA cells were killed by AuNPs@LL37 (15 μg mL^−1^, corresponding to an immobilized concentration of 0.86 μg mL^−1^ LL37) and AuNPs@LL37/pDNAs (15 μg mL^−1^) after incubation for 4 h.	[97]
	(4) Small synthetic peptide bioconjugated to gybrid gold nanoparticles 1018K6-AuNPs with an average size of 14 ± 7 nm.	*Listeria Monocytogenes* and *Salmonella Typhimurium*	1018K6 of a concentration of <100 nM tethered on AuNPs eliminated almost 100% of treated pathogen bacteria, such as *Listeria* and *Salmonella* genera.	[98]
Gold nanoparticles conjugated to enzymes	(1) Ampicillin (Amp) conjugated to lysozyme-capped gold nanoclusters (AUNCs) to form the conjugate AUNC-L-Amp with an average size of 2.71 ± 0.15 nm.	*E. coli*, *S. aureus, S. epidermidis*, *Bacillus subtilis*, *Bacillus cereus, Micrococcus luteus*, *Klebsiella pneumoniae*, *P. aeruginosa*, *Proteus vulgaris,* and MRSA	50–89% fold increase in antibacterial activity of AUNC-L-Amp compared to free-ampicillin against 9 nonresistant bacterial pathogens, and enhanced activity against 10 MRSA clinical isolates, in comparison to free-amp.	[99]
	(2) Protease-conjugated gold nanorods (PGs) of an average length of 32 nm and an average width of 7.8 nm.	*E. coli* and *S. aureus*	Using this PGs system, the bacterial survival rate population was reduced to 3.2% and 2.1% of untreated control numbers for *E. coli* and *S. aureus*, respectively.	[100]
Theranostic antibacterial systems based on gold nanoparticles	Tetrazine-modified gold nanoparticle (GNP-Tz) of an overall diameter of 25 ± 5 nm.	*Bacillus subtilis*, *S. aureus*, *Enterococcus faecalis,* and *E. coli*	More than 90% of the Gram-positive bacterial cells were dead under 10 min NIR irradiation, while for Gram-negative *E. coli* the system was not effective.	[102]
Pre-treated macrophage-membrane-coated gold–silver nanoparticles	Pretreated macrophage membranes where tethered to gold–silver nanocages (GSNCs) with a hydrodynamic diameter of ~125 nm.	*S. aureus*	The *S. aureus* macrophage treated gold–silver nanocage Sa-M-GSNC nanoconjugate completely inhibited bacterial growth within the first 6 h, with laser irradiation.	[103]
Gold nanosystems targeting bacterial biofilms	(1) Deoxyribonuclease (DNase)-functionalized gold nanoclusters (AuNCs) of spherical shapes with an average diameter of ~2.3 nm.	Multidrug-resistant (MDR) *S. aureus* or MDR *P. aeruginosa*	The nanosystem was capable of removing biofilms with a dispersion rate of up to 80% and kill ∼90% of the shielded bacteria.	[107]
	(2) Laser-induced vapor nanobubbles (VNBs) by using cationic 70-nm gold nanoparticles.	Biofilms of Gram-negative *Burkholderia multivorans*, *P. aeruginosa*, and Gram-positive *S. aureu*s	In all types of biofilms tested, tobramycin efficacy increased up to 1–3 orders of magnitude depending on the organism and treatment conditions.	[108]
Gold nanoparticles conjugated to protein (BSA)	Bovine serum albumin (BSA) and 4,6-diamino-2-pyrimidinethiol (DAPT) conjugated AuNPs of an indefinite shape with a diameter of 4.11 ± 0.32 nm.	*P. aeruginosa*, *S. aureus*, *E. coli, Acinetobacter baumannii, Klebsiella oxytoca*, *Serratia marcescens*, *Enterobacter cloacae*, *Burkholderia cepacian, E. faecium, Streptococcus dysgalactiae*, *Streptococcus agalactiae*, *Enterococcus faecalis*, and *Streptococcus pyogenes*	MIC less than < 16 μg/mL. MDR strains can be 99.9% eliminated after incubating bacteria and Au_DAPT_BSA for 12 h, in which the concentration of Au_DAPT_BSA is 1 and 32 μg/mL for MDR *E. coli* and MRSA, respectively.	[111]
Gold nanoparticles attached to aminosacharrides D-glucosamine (GluN), D-galactosamine (GalN), or D-mannosamine (ManN)	AuNPs of an indefinite shape and around ~4 nm in diameter.	*S. aureus, S. epidermidis, Listeria monocytogenes, B. subtilis, E. faecium,* MRSA, and MDR *S. epidermidis*	Au_GluN conjugate with the ratio of Au:GluN at 1.00:0.42 showed the best antibacterial activity with MIC of <4 μg/mL.	[112]
Targeted therapy for gastrointestinal bacteria by D-Au NPs	DAPT-coated Au nanoparticles (D-Au NPs) of a spherical shape ~5 nm in diameter.	*E. coli*	D-Au NPs at a concentration 34 μg/mL were capable of halting bacterial growth of *E. coli* for 72 h.	[114]

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
