# Peer review of "Gold Nanoparticles: Can They Be the Next Magic Bullet for Multidrug-Resistant Bacteria?"

_nanomaterials, 2021, doi:10.3390/nano11020312_

Round 1

Reviewer 1 Report

Visai et al present, through a nicely suggestive title, the most important results regarding the use of gold nanoporticles to combat multidrug-resistant bacteria. The description of the results, along the multiple approaches involving AuNPs, is comprehensive and clear, thus confidently deserving their publication in Nanomaterials.

The minor point that I would suggest, for a much clearer picture of the most important results, is the inclusion of a collective table in order to gather, per each type of approach, the results of antibacterial action of AuNPs in terms of size, morphology, bacteria type, antibacterial efficcacy, and related references.

Author Response

Reviewer 1 - Visai et al present, through a nicely suggestive title, the most important results regarding the use of gold nanoporticles to combat multidrug-resistant bacteria. The description of the results, along the multiple approaches involving AuNPs, is comprehensive and clear, thus confidently deserving their publication in Nanomaterials.
The minor point that I would suggest, for a much clearer picture of the most important results, is the inclusion of a collective table in order to gather, per each type of approach, the results of antibacterial action of AuNPs in terms of size, morphology, bacteria type, antibacterial efficacy, and related references.

We thank you the reviewer for the positive comment.

The manuscript was improved with the addition of Table 2, containing the antibacterial action of AuNPs in terms of size, morphology, bacteria type, antibacterial efficacy, and related references. 

Reviewer 2 Report

Regarding to era of antimicrobial resistance this manuscript is very interesting and actual. Authors try to present detail characterisation of gold nanoparticles and their application. However, to improve this work, I have a few suggestions:

  • In the first issue, moreinformation about epidemiology of bacterial infection based on actual database should be provided.
  • A table with more example of application of gold nanoparticles against microbial treatment should be added
  • In toxicological section there is lack information about pharmacological properties of gold NPs some references might be included:

Xia Q, Li H, Xiao K. Factors Affecting the Pharmacokinetics, Biodistribution and Toxicity of Gold Nanoparticles in Drug Delivery. Curr Drug Metab. 2016;17(9):849-861. doi: 10.2174/1389200217666160629114941. PMID: 27364829

Schmid G, Kreyling WG, Simon U. Toxic effects and biodistribution of ultrasmall gold nanoparticles. Arch Toxicol. 2017 Sep;91(9):3011-3037. doi: 10.1007/s00204-017-2016-8. Epub 2017 Jul 12. PMID: 28702691.

Riviere JE, Jaberi-Douraki M, Lillich J, Azizi T, Joo H, Choi K, Thakkar R, Monteiro-Riviere NA. Modeling gold nanoparticle biodistribution after arterial infusion into perfused tissue: effects of surface coating, size and protein corona. Nanotoxicology. 2018 Dec;12(10):1093-1112. doi: 10.1080/17435390.2018.1476986. Epub 2018 Jun 1. PMID: 29856247.

etc.

Author Response

Reviewer 2 - Regarding to era of antimicrobial resistance this manuscript is very interesting and actual. Authors try to present detail characterisation of gold nanoparticles and their application. However, to improve this work, I have a few suggestions:

  • In the first issue, more information about epidemiology of bacterial infection based on actual database should be provided.
  • A table with more example of application of gold nanoparticles against microbial treatment should be added
  • In toxicological section there is lack information about pharmacological properties of gold NPs some references might be included:

Xia Q, Li H, Xiao K. Factors Affecting the Pharmacokinetics, Biodistribution and Toxicity of Gold Nanoparticles in Drug Delivery. Curr Drug Metab. 2016;17(9):849-861. doi: 10.2174/1389200217666160629114941. PMID: 27364829

Schmid G, Kreyling WG, Simon U. Toxic effects and biodistribution of ultrasmall gold nanoparticles. Arch Toxicol. 2017 Sep;91(9):3011-3037. doi: 10.1007/s00204-017-2016-8. Epub 2017 Jul 12. PMID: 28702691.

Riviere JE, Jaberi-Douraki M, Lillich J, Azizi T, Joo H, Choi K, Thakkar R, Monteiro-Riviere NA. Modeling gold nanoparticle biodistribution after arterial infusion into perfused tissue: effects of surface coating, size and protein corona. Nanotoxicology. 2018 Dec;12(10):1093-1112. doi: 10.1080/17435390.2018.1476986. Epub 2018 Jun 1. PMID: 29856247.

We thanks the reviewer for the positive comment.

Regarding the indicated points we impoved the manuscript as follows:

-A table 1 was added with the information about epidemiology of bacterial infection based on actual database

-A table 2 was added with the information regarding the application of gold nanoparticles against microbial treatments

- The toxicological section was completely revised with information about the pharmacological properties of gold NPs including the indicated references.